# Polymicrobial PID Presenting as Primary Peritonitis in a Young Immunocompetent Patient—Case Report and Disease Perspectives

**DOI:** 10.3390/diagnostics16010134

**Published:** 2026-01-01

**Authors:** Georgiana Nemeti, Maria Adriana Neag, Iulian Gabriel Goidescu, Mihai Surcel, Cerasela Mihaela Goidescu, Ioana Cristina Rotar, Daniel Muresan

**Affiliations:** 1Obstetrics and Gynaecology I, Mother and Child Department, “Iuliu Hatieganu” University of Medicine and Pharmacy, 400006 Cluj-Napoca, Romania; georgiana.nemeti@elearn.umfcluj.ro (G.N.); mihai.surcel@elearn.umfcluj.ro (M.S.); cristina.rotar@umfcluj.ro (I.C.R.); muresandaniel01@elearn.umfcluj.ro (D.M.); 2Department of Pharmacology, Toxicology and Clinical Pharmacology, “Iuliu Hatieganu” University of Medicine and Pharmacy, 400337 Cluj-Napoca, Romania; maria.neag@elearn.umfcluj.ro; 3Department of Internal Medicine, Medical Clinic I—Internal Medicine, Cardiology and Gastroenterology, “Iuliu Hatieganu” University of Medicine and Pharmacy, 400006 Cluj-Napoca, Romania; sava.cerasela@elearn.umfcluj.ro

**Keywords:** PID, primary peritonitis, sexually transmitted infections, immunocompetent, *Chlamydia trachomatis*, *Mycoplasma hominis*, *Ureaplasma parvum*

## Abstract

**Background and Clinical Significance**: Pelvic inflammatory disease represents a multifaceted sexually transmitted disease affecting women of reproductive age, beginning in adolescence. Clinical presentation ranges from asymptomatic patients to acute abdominal pain in the setting of tubo-ovarian abscesses; however, presentation as primary peritonitis with seemingly intact fallopian tubes is exceptional. Primary peritonitis in the absence of other comorbid conditions (e.g., liver cirrhosis and nephrotic syndrome) in healthy, immunocompetent women is rare and typically occurs without an identifiable intra-abdominal source. The diagnosis can be challenging due to its mild-to-moderate, nonspecific symptoms. **Case Presentation:** We report the case of a 21-year-old immunocompetent woman who presented with lower abdominal and left iliac fossa pain with hyperleukocytosis. Laparoscopic exploration confirmed the diagnosis of primary peritonitis. Following diagnosis, she underwent peritoneal lavage and was started on empiric broad-spectrum parenteral antibiotic therapy. Cervico-vaginal cultures established the diagnosis of PID following identification of *Chlamydia trachomatis*, *Mycoplasma hominis*, and *Ureaplasma parvum*. The clinical course was favorable. **Conclusions:** An early multidisciplinary approach, including consultation with an infectious disease specialist and clinical pharmacologist, is recommended in cases of peritonitis with an unclear source. PID may present as primary peritonitis and this clinical scenario should be considered in sexually active young women with unexplained peritoneal infection when no gastrointestinal or gynecologic source is evident intraoperatively.

## 1. Introduction

Pelvic inflammatory disease (PID) is a polymicrobial infection that predominantly affects sexually active young cisgender women, but also women of older reproductive age [1]. Upper genital tract inflammatory processes lead to a broad spectrum of conditions, such as acute salpingitis, perihepatitis, endometritis, oophoritis, tubo-ovarian abscess, and pelvic peritonitis [2,3], most commonly caused by the presence of *Chlamydia trachomatis* and/or *Neisseria gonorrhoeae* [4]. Frequently, the infection becomes polymicrobial, potentiated by the presence of various vaginal–cervical microorganisms [5,6].

Patients may be asymptomatic, with incidental retrospective diagnosis in suggestive clinical contexts such as surgery for complicated tubal pregnancy, infertility workup, or during pelvic surgery for other indications. Symptoms range from lower abdominal pain, pelvic organ tenderness, and purulent vaginal discharge, and may have an insidious or abrupt onset.

The population most susceptible to PID are adolescents and young females given their topical cervico-vaginal particularities—cervical ectropion and an immature immune system (both local and systemic)—as well as frequently reckless sexual behavior (multiple coital partners, use of non-barrier contraception, ineffective condom usage). Additional risk factors include reproductive age patients with bacterial vaginosis, vaginal douching, coitus during menstruation, and a history of PID episodes (which often lack proper diagnosis and management) [7].

Careful evaluation of PID, especially in the setting of acute presentations, is mandatory to achieve a correct diagnosis and ensure proper and timely management to prevent the development of sequalae.

Primary peritonitis refers to peritoneal inflammation and infection in the absence of any identifiable lesions in the gastrointestinal or genitourinary tracts, or in solid organs [8]. Although its pathogenesis is not fully understood, several mechanisms have been proposed, including bacterial translocation from the intestine, hematogenous dissemination, and retrograde spread from the genitourinary tract [9].

*Chlamydia trachomatis*, *Mycoplasma hominis*, and *Ureaplasma parvum* are part of the genital microbiota and fall under the category of sexually transmitted infections (STIs). These organisms typically infect epithelial cells and rarely enter the bloodstream [10,11]. *Chlamydia trachomatis* is the most prevalent bacterial agent involved in STIs and should always be considered in cases of pelvic inflammatory disease [12]. *Mycoplasma hominis* also plays a significant role in female reproductive tract inflammation, often in co-infection with *Chlamydia*, *Trichomonas vaginalis*, or *Neisseria gonorrhoeae*, presenting with similar clinical features [10,11,12,13]. *Ureaplasma parvum* is generally associated with asymptomatic infections, but chronic colonization may contribute to upper genital tract inflammation and, in rare cases, peritonitis [14,15].

In this case report, we present an unusual case of PID manifesting as primary peritonitis with abrupt onset and rapid worsening in a healthy, sexually active young woman, with a polymicrobial sexually transmitted infection panel positive for *Chlamydia trachomatis*, *Mycoplasma hominis*, and *Ureaplasma parvum*. Although polymicrobial PID is common, initial presentation as an acute surgical abdomen with primary peritonitis in the absence of macro-lesional tubal damage is exceptional, and even more so in an immunocompetent young patient. This raises awareness regarding the importance of considering such etiologies in the differential diagnosis of acute abdomen, particularly in sexually active patients without identifiable intra-abdominal pathology.

## 2. Case Presentation

A 21-year-old healthy woman, without relevant obstetrical–gynecologic, medical, or surgical history, with regular menstrual cycles, currently in the periovulatory period, was admitted to our department through the emergency room for pain located in the lower abdomen and left iliac fossa. The pain was intense, with a VAS (Visual Analog Scale) score of 7–8, and persisted regardless of the antalgic and anti-inflammatory medication provided.

The patient was afebrile (temperature 36.6 °C), her blood pressure was 130/60 mmHg, with a normal pulse. Clinical examination revealed a soft and flat abdomen, with spontaneous and palpable pain and tenderness in the left iliac fossa and hypogastrium. The bimanual pelvic examination objectified acute cervical motion, uterine, and adnexal tenderness. Transvaginal ultrasonographic examination revealed a minimal amount of free fluid in the pouch of Douglas, consistent with ovulation, but otherwise normal internal genitalia, confirming the periovulatory stage, and no other pathologic findings of the surrounding organs. Based on clinical and imaging findings, an initial diagnosis of uncomplicated PID was established, and the patient was admitted for surveillance and management. A vaginal discharge slide exam and a swab for sexually transmitted infections (STIs) screening were sampled (NAAT—nucleic acid amplification test). Empiric broad-spectrum antibiotic therapy was initiated, consisting of clindamycin, gentamicin, and ceftriaxone, along with antifungal protection, anti-algic, and anti-inflammatory therapy, gastric protection, and fluid balancing solutions. Initial lab workup showed a leukocyte count of 21,000 × 10^9^/L, a C-reactive protein level of 16.67 mg/dL, a fibrinogen value of 681 mg/dL, a ferritin value of 147 ng/mL, and mildly elevated direct and indirect bilirubin, all supporting an inflammatory state with a mild hepatic response.

Approximately 5 h later, the pain intensified, and upon ultrasound reevaluation, an increased quantity of stained free fluid were identified by endovaginal ultrasound examination in the Douglas pouch and ante-uterine space, which we interpreted as peritonitis. Surgical evaluation was requested, and the common diagnostic achieved was acute surgical abdomen. Given the severe abdominal pain intensifying despite the medication provided, as well as the increasing amount of impure peritoneal fluid, following patient consent and perioperative preparations, an exploratory laparoscopy was performed. At initial examination, approximately 200 mL of purulent effusion was observed in the peritoneal cavity (Figure 1), with an unremarkable-looking uterus, ovaries, and fallopian tubes. There were no signs of adhesions, tubal congestion, enlargement, or purulent discharge coming from the salpinges at the time of exploration; both tubes depicted a normal course. Bacteriological samples were obtained, and abundant peritoneal lavage was carried out. The appendix, intestines, and gallbladder were normal, and no perforation of other pelvic/abdominal organs could be identified. In the absence of a clear source, the diagnosis of primary purulent peritonitis was established, and broad-spectrum antibiotic therapy with piperacillin/tazobactam and doxycycline was started.

Bacteriological samples were negative, but PCR/NAAT (polymerase chain reaction/nucleic acid amplification testing) was not available in the emergency setting. However, the sexually transmitted disease panel from vaginal and cervical secretions was positive for *Chlamydia trachomatis*, *Mycoplasma hominis*, and *Ureaplasma parvum* (Table 1). The wet prep revealed abundant leukocytes. Further STI testing for hepatitis B and C, HIV (human immunodeficiency virus), and syphilis were negative.

The postoperative course of the patient was uneventful, with same-day mobilization and transit restoration, as well as a rapid decrease in inflammatory response parameters under the treatment regimen (Table 2). Hepato-protective therapy was initiated due to a mild hepatic reaction. The patient was discharged on day 4 with a doxycycline and amoxicillin–clavulanate antibiotic regimen to be carried out until 14 days of treatment would be completed, as well as single-dose azithromycin therapy for the sexual partner. The choice of the outpatient antibiotic regimen was motivated by a step-down approach while maintaining the antimicrobial profile and the need to cover gut flora translocation despite negative cultures. Extensive patient counseling regarding sexual hygiene, PID episodes prophylaxis, reevaluation schedule, as well as potential long-time sequalae was provided. At 21 days post-surgery, the patient presented for follow-up and was asymptomatic, with a normal pelvic ultrasound and a normal blood test panel. The patient then presented for bi-annual follow-ups, which were unremarkable.

## 3. Discussion

Pelvic inflammatory disease represents the consequence of ascending pathogen dissemination from the lower to the upper genital tract and the peritoneal cavity, sometimes affecting neighboring pelvic organs (bowel, omentum, bladder).

The hallmark of the disease is pelvic pain and tenderness. In the initial stages of the disease, when tubal compromise is subtle and there are no imaging signs of damage, PID is a diagnosis of exclusion in the context of lower abdominal pain and tenderness, absence of other identifiable genital tract, pelvic, or abdominal organ pathology, and confirmation by positive testing for specific sexually transmitted bacteria.

The most commonly identified agents involved in PID are Chlamydia trachomatis, isolated in 10–43% of cases, and Neisseria gonorrhoeae, found in 25–50% of patients [4,7,16,17]. In 30% of cases, other microorganisms are involved, such as bacterial vaginosis agents, cervical pathogens, enteric bacteria, respiratory microbes, and other anaerobic and facultative species (Table 3). Many cases depict polymicrobial infections; contrarily, in an important share of patients with macro-lesional disease and even purulent vaginal discharge/purulent peritoneal fluid, an etiologic agent is not identified [16].

Factors favoring the occurrence of PID include young age, early age of sexual debut, an immature immune system, an immature cervical tissue represented by columnar epithelium in the transitional zone, which maintains a propitious ambient for the growth of Chlamydia trachomatis and Neisseria gonorrhoeae, multiple sex partners, unhealthy sexual behaviors (inconsistent condom use, contacts during menses, non-barrier contraceptive means), vaginal douching, history of STIs, previous PID episodes, intra-uterine device use (recent insertion, extended use), recent oocyte retrieval or pelvic surgery, and recent endometrial biopsy [18,19]. In many cases of PID, however, no risk factors can be identified. In consistency with literature reports, our patient was young but had no positive history of previous PID episodes, endo-uterine maneuvers, or risky sexual habits.

Patients with PID have varying symptomatology ranging from subtle pain to acute surgical abdomen, often making diagnosis difficult. The presence of sactosalpinx or tubo-ovarian abscesses, with or without pelvic collections, calls for a swift diagnosis. In cases where there is no clear image of adnexal involvement, diagnosis becomes more challenging [20].

Peritonitis is a life-threatening acute surgical emergency requiring prompt intensive care and surgical management and may be classified into three categories: primary (or spontaneous) peritonitis, which occurs in the absence of any intra-abdominal surgical focus; secondary peritonitis, resulting from the loss of integrity of an intra-abdominal organ or a penetrating infectious process; and tertiary peritonitis, defined by the persistence or recurrence of signs and symptoms of peritonitis following adequate treatment of secondary peritonitis [21]. Primary peritonitis is exceptionally rare in young immunocompetent patients, accounting for less than 1% of all peritonitis cases [22].

The case of our patient can be regarded from two equally valid perspectives. On the one hand, as we initially thought, in the context of abdominal pain and pelvic examination tenderness without suggestive imaging findings for intra-abdominal pathology, it can be regarded as uncomplicated PID. As the clinical evolution of the patient swiftly developed towards worsening general status and increasing stained peritoneal fluid collection, in the absence of microbiological confirmation, we then interpreted it as primary peritonitis until cervical test results were available. In reality, rapid development of PID to peritonitis within hours is rare, especially in the absence of long-standing disease and established tubal damage.

The initial treatment regimen chosen was empirical, consistent with the diagnosis of PID, and aimed to cover the most common STI spectrum, associating clindamycin, gentamicin, and ceftriaxone. Timely and correct antibiotic therapy is meant to prevent chronic sequelae such as adhesions, scarring, tubal damage, ectopic pregnancy, infertility, and chronic pelvic pain [20,23]. Following the arrival of culture results confirming the diagnosis of STI, the antimicrobial regimen was adjusted to include piperacillin/tazobactam, a broad-spectrum beta-lactam/beta-lactamase inhibitor combination effective against a wide range of aerobic and anaerobic bacteria. Due to the absence of a peptidoglycan cell wall, *Mycoplasma hominis* is intrinsically resistant to most broad-spectrum antibiotics, including penicillins, cephalosporins, and carbapenems. Recommended treatment options include tetracyclines or fluoroquinolones [24]. Therefore, doxycycline, a tetracycline-class antibiotic, was added to the therapeutic protocol because it is active against *Chlamydia trachomatis*, *Mycoplasma hominis*, and *Ureaplasma parvum* and is considered a first-line agent in the treatment of STIs involving these pathogens. This combined approach ensured both empirical coverage of potential intra-abdominal pathogens and targeted therapy for the identified agents.

To place our case in a broader clinical context, from the perspective of primary peritonitis, Table 4 summarizes previously reported cases and studies involving the main pathogenic agents encountered. It includes key patient characteristics, identified infectious agents, diagnostic methods, and therapeutic approaches. It also highlights the use of empiric broad-spectrum antibiotic regimens, often initiated before pathogen identification and subsequent targeted therapies administered based on microbiological findings.

The exact mechanism through which these genital pathogens reached the peritoneal cavity in our patient remains unclear. However, an ascending infection pathway from the lower genital tract represents the most plausible explanation, especially in the absence of any other identifiable intra-abdominal source, regardless of whether it belongs to the category of STIs. This mechanism has been previously described in cases of pelvic inflammatory disease progressing to pelvic or generalized peritonitis, particularly when infection is left untreated or inadequately treated. However, if this had been the case in our patient, she should either have reported previous episodes of strong abdominal pain, or tubal compromise might have been apparent.

Our patient presented a favorable clinical outcome, even though treatment was empirical due to initially negative culture results. This may be attributed to the fastidious natures of Mycoplasma hominis and Ureaplasma parvum, which require special culture media and conditions not routinely used in standard bacterial cultures. Additionally, prior initiation of antibiotic therapy before surgical sampling could have further reduced the likelihood of isolating viable organisms.

Chienwichai et al. suggest that culture-negative peritonitis is often associated with more favorable clinical outcomes compared to culture-positive cases, despite ongoing uncertainty surrounding its underlying mechanisms [33]. Notably, hypokalemia emerged as a potential predictor of culture-negative peritonitis in that cohort. A plausible mechanism may explain this association—hypokalemi-induced increases in intestinal permeability may facilitate bacterial translocation, particularly of anaerobic and Gram-negative organisms that may evade detection in standard cultures. This factor may contribute to the higher prevalence of hypokalemia observed in culture-negative cases and highlight the need for further investigation into the pathophysiology and diagnostic limitations associated with this condition [33].

*Chlamydia trachomatis* is the most widespread STI pathogen and most frequently induces a silent but persistent infection leading to inflammation and chronic injury of genital tract tissues [6,7].

*Mycoplasma hominis* is strongly associated with bacterial vaginosis and, to a lesser extent, with PID. Genital mycoplasmas may be a source of systemic infections, often presenting with sepsis, with *Mycoplasma hominis* being the most commonly associated pathogen [34,35].

*Ureaplasma parvum* and *Ureaplasma urealyticum* are part of the human genital tract microbiota. Distinguishing between these sub-species prevents unnecessary treatment of *Ureaplasma parvum*, which is generally considered a colonizing pathogen rather than a causative one [36]. However, in a report of ten cases of peritonitis due to *Ureaplasma*, six were due to the *parvum* sub-type [11].

Even though *Mycoplasma hominis* and *Ureaplasma parvum* agents are not reported as the most aggressive PID etiologies, in our case, it appears that their association, together with *Chlamydia trachomatis*, led to a rapidly progressive disease.

An early multidisciplinary approach, including consultation with an infectious disease specialist and a clinical pharmacologist, is recommended in cases of peritonitis with an unclear source. Awareness must also be maintained among both gynecologists and general surgeons regarding the potential first presentation of PID as primary peritonitis.

Female patients presenting with PID, regardless of the type of presentation, must receive counseling regarding long-time sequalae which may be a consequence of the long-standing or recurrent inflammatory processes, such as adhesions, scarring, tubal obliteration, traction, and functional impairment. These changes favor tubal ectopic implantation of embryos, lead to infertility, and predispose patients to chronic pelvic pain [37,38]. More recent studies have investigated the potential relationship between PID and subsequent development of endometriosis, with unclear results [39,40], while others have explored the link between PID and genital cancers—ovarian, cervical, and uterine [41,42,43,44,45].

Prophylactic measures, including efforts to increase the level of awareness among adolescents and young adults regarding STI protection, PID symptoms, and methods to avoid infections or attend physicians for a timely diagnosis and correct management, are mandatory.

## 4. Conclusions

In conclusion, although rare, primary peritonitis caused by genital tract pathogens in the form of acute PID must be considered in sexually active young women with unexplained peritoneal infection, particularly when conventional cultures are negative and no gastrointestinal or gynecologic source is evident intraoperatively.

## Figures and Tables

**Figure 1 diagnostics-16-00134-f001:**
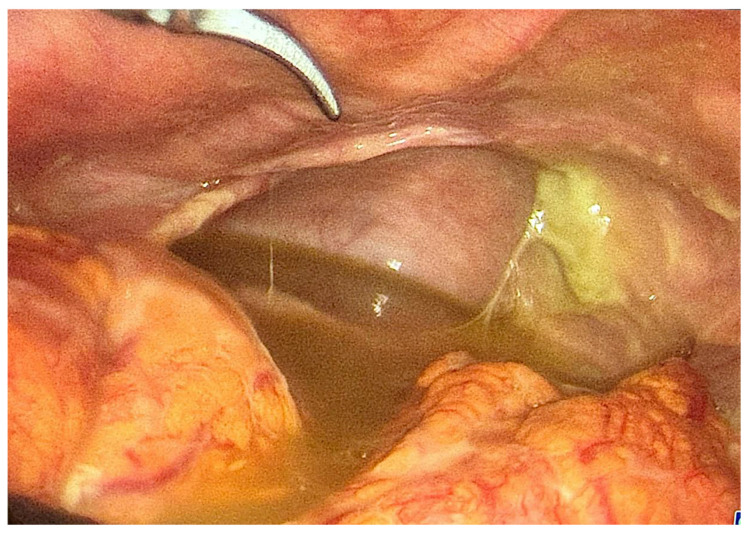
Initial aspect of pelvic cavity at laparoscopic entry with a significant amount of purulent collection.

**Table 1 diagnostics-16-00134-t001:** Patient STI infection panel result.

Sexually Transmitted Diseases Panel—Nucleic Acid Amplification Testing (NAAT) from Cervico-Vaginal Swab Collection		
Pathogenic Agent	Patient Results	Reference Value
*Chlamydia trachomatis*	**positive**	negative
*Mycoplasma hominis*	**positive**	negative
*Ureaplasma parvum*	**positive**	negative
*Neisseria gonorrhoeae*	negative	negative
*Mycoplasma genitalium*	negative	negative
*Ureaplasma urealyticum*	negative	negative
*Trichomonas vaginalis*	negative	negative

**Table 2 diagnostics-16-00134-t002:** Blood test results of patients during the admission timeframe (red—increased values; blue—decreased values).

Blood Test	Value Day 1	Value Day 2	Value Day 4	Value Day 6	Reference Ranges	Units
White blood cell count	21.45	24.83	9.97	9.47	4–10	(10^3^/μL)
Red blood cell count	4.41	3.83	3.61	3.92	4–5	(10^4^/μL)
Hemoglobin	13.1	11.6	10.6	11.4	12.0–15.5	(g/dL)
Hematocrit	39.8	34.7	32.8	34.8	37–47	(%)
Platelet count	357	318	351	414	150–400	(10^4^/μL)
Prothrombin index	77			82	80–151	(%)
Activated partial thromboplastin time	36.3			42.6	23.5–36.5	(s)
Sodium	134		142	140	136–146	(mmol/L)
Potassium	3.7		3.49	3.95	3.5–5.1	(mmol/L)
Blood urea nitrogen	24		30	20	17–43	(mg/dL)
Creatinine	0.78		1.16	0.82	0.51–0.95	(mg/dL)
Total bilirubin	1.31		0.2	0.31	0.3–1.2	(mg/dL)
Albumin				3.6	3.4–5.4	(g/dL)
AST	22		26	96	<35	(IU/L)
ALT	10		20	98	<35	(IU/L)
Lactate dehydrogenase				281	<247	(IU/L)
Amylase					25–110	(IU/L)
C-reactive protein	16.67	18.85	4.63	2.18	<0.5	(mg/dL)
Procalcitonin		0.556	0.187	0.062	0.1	(ng/mL)
Ferritin	147			108	10–120	(ng/mL)
Fibrinogen	681				200–400	(mg/dL)

AST: aspartate aminotransferase; ALT: alanine aminotransferase; NAAT: nucleic acid amplification test.

**Table 3 diagnostics-16-00134-t003:** Microorganisms isolated from PID patients.

Microorganisms Involved in PID	
Most Frequently Identified	Other Microorganisms
*Chlamydia trachomatis* *Neisseria gonorrhoeae* *Mycoplasma genitalium*	*Mycoplasma hominis* *Ureaplasma urealyticum* *Ureaplasma parvum* *Neisseria meningitides* *Gardnerella vaginalis* *Group B Streptococcus* (*S. agalactiae*) *Bacteroides species* (*fragilis*, *bivius*, *disiens*) *Streptococcus faecalis* *Haemophilus influenzae* *Coliforms* (*Enterobacteriaceae*) *Enterococcus* *Cytomegalovirus* *Peptostreptococcus* *Other anaerobes*

**Table 4 diagnostics-16-00134-t004:** Primary peritonitis cases previously reported in the literature in both female and male patients.

Study	Patient Characteristics (Gender, Age, and Special Mentions)	Peritoneal Fluid Microbiologic Samples	Treatment
Rai, 2014 [25]	Female 43 years	*Klebsiella pneumoniae*	- empiric ampicillin, gentamicin, and metronidazole - oral cefalexin
Blevrakis, 2016 [26]	Female 14 years Mild upper respiratory infection two weeks ago	*Streptococcus pneumoniae* (serotype 3)	- empiric ceftazidime and metronidazole - ceftriaxone
Drexel, 2018 [24]	Female 42 years Uterine fibroids	Gram stain—negative 16S ribosome testing—*Mycoplasma hominis* RNA	- empiric antibiotics for PID (ceftriaxone, azithromycin, and metronidazole IV) - doxycycline p.o.
Ugrinovic, 2020 [27]	Female 16 years Presumptive diagnosis—appendicitis	*Streptococcus pneumoniae*	- piperacillin–tazobactam
Sumiyama, 2022 [28]	Female 56 years Abnormal vaginal discharge for two weeks	*Group A Streptococcus*	- empiric meropenem for six days - levofloxacin orally
Petri, 2024 [11]	Female 36 years History of metastatic sigmoid colon adenocarcinoma	*Ureaplasma parvum*	- empiric piperacillin/tazobactam, later with caspofungin and vancomycin add-on - doxycycline orally
Matsumoto, 2024 [29]	Female 42 years Cesarean section three months prior	*Group A Streptococcus*	- empiric piperacillin/tazobactam - penicillin G and clindamycin
Barrés-Fernández, 2020 [30]	Male 11 years Initial diagnosis—peritonitis and pleural effusion Final—Ménétrier’s Disease	Negative	- empiric—cefotaxime - empiric—vancomycin was added
Antonik, 2021 [31]	Male 49 years	*Chlamydia trachomatis* Syphilis serology positive	- oral doxycycline - single-dose of benzathine penicillin
Gizzatullin, 2024 [32]	Male 35 years	*Group A Streptococcus*	- empiric cefuroxime and metronidazole - amoxicillin–clavulanate and clindamycin

## Data Availability

The original contributions presented in this study are included in the article. Further inquiries can be directed to the corresponding author.

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
