# Peer review of "Diagnostics2026, 16(1), 134;https://doi.org/10.3390/diagnostics16010134"

_diagnostics, 2026, doi:10.3390/diagnostics16010134_

Round 1

Reviewer 1 Report

Comments and Suggestions for Authors

The authors present an interesting and well-documented case of primary purulent peritonitis in a young immunocompetent woman, associated with a polymicrobial sexually transmitted infection panel (Chlamydia trachomatis, Mycoplasma hominis and Ureaplasma parvum).

The topic is clinically relevant and fits the scope of the journal, particularly given the rarity of primary peritonitis in healthy patients and the diagnostic challenges highlighted.

The manuscript is well written, and the clinical course is clearly described. Imaging, laparoscopic findings, laboratory data and follow-up are appropriately reported. The inclusion of a literature review table adds value and contextualizes the case within previously reported presentations.

However, I have some concerns about the paper:

  1. Discussion needs expansion: the current discussion is adequate but remains descriptive. The authors should further elaborate on:

    • the proposed mechanisms of ascending infection leading to peritoneal involvement in the absence of pelvic inflammatory findings;

    • the diagnostic rationale for proceeding to laparoscopy after 5 hours and how this aligns with current recommendations;

    • the significance of culture-negative peritoneal fluid in the context of Mycoplasma/Ureaplasma infections and limitations of standard culture techniques.

  2. Clarity in Table 1: consider adding reference ranges to improve readability. 

  3. Language and style: the English language is generally good, but minor grammatical and stylistic corrections are recommended to improve fluency and precision.

  4. The similarity analysis indicates a 19% match: I encourage the authors to carefully review the segments highlighted by the plagiarism checker. In several parts, particularly within the introduction and discussion, the similarity may derive from previously published reviews or guidelines.

With these revisions, the manuscript will be strengthened and suitable for publication.

Author Response

Esteemed Reviewer, thank you taking the time to read and evaluate our work, for your appreciation of our manuscript and for outlining our shortcomings. We have done our best to ammend the article, you were quite right on every aspect and we are thankful for your suggestions. All changes brought to the manuscript are highlighted in blue.

Reviewer: Discussion needs expansion: the current discussion is adequate but remains descriptive. The authors should further elaborate on:

    • the proposed mechanisms of ascending infection leading to peritoneal involvement in the absence of pelvic inflammatory findings;
    • the diagnostic rationale for proceeding to laparoscopy after 5 hours and how this aligns with current recommendations;
    • the significance of culture-negative peritoneal fluid in the context of Mycoplasma/Ureaplasma infections and limitations of standard culture techniques.

Authors: Thank you for your observations.

  • Regarding the ascending infection mechanism of the declared pathoges, it has already been stated and proven more soundly by many authors. It is true we did not have positive peritoneal fluid culture because the setup of the hospital did not allow us to perform STI twice in the same patient, regardless it would have been from two collection sites. At the same time, this is the drive behind our case presentation – the strangeness of normally looking, feeling, normal anatomical course tubes, the presence of peritonitis, and the positive cervico-vaginal PCR.
  • We have ammended the case presentation to state more clearly the rationale behind the decision to operate. The patient’s abdominal pain intensified despite medical management and the peritoneal fluid quantity augmented (stained, puss-like fluid).

Reviewer: Clarity in Table 1: consider adding reference ranges to improve readability. 

Authors: Thank you for your suggestion, we have added them.

Reviewers: Language and style: the English language is generally good, but minor grammatical and stylistic corrections are recommended to improve fluency and precision.

Authors: Thank you for your obervation. We have re-read and reshaped much of the article, we hope it sounds better now and no mistakes have escaped our reading.

Reviewer: The similarity analysis indicates a 19% match: I encourage the authors to carefully review the segments highlighted by the plagiarism checker. In several parts, particularly within the introduction and discussion, the similarity may derive from previously published reviews or guidelines.

With these revisions, the manuscript will be strengthened and suitable for publication.

Authors: Thank you for poiting this out, we did not copy-paste from other sources into our work. I would say there is little literature on

Esteemed Reviewer, thank you taking the time to read and evaluate our work, for your appreciation of our manuscript and for outlining our shortcomings. We have done our best to ammend the article, you were quite right on every aspect and we are thankful for your suggestions. All changes brought to the manuscript are highlighted in blue.

Reviewer: Discussion needs expansion: the current discussion is adequate but remains descriptive. The authors should further elaborate on:

    • the proposed mechanisms of ascending infection leading to peritoneal involvement in the absence of pelvic inflammatory findings;
    • the diagnostic rationale for proceeding to laparoscopy after 5 hours and how this aligns with current recommendations;
    • the significance of culture-negative peritoneal fluid in the context of Mycoplasma/Ureaplasma infections and limitations of standard culture techniques.

Authors: Thank you for your observations.

  • Regarding the ascending infection mechanism of the declared pathoges, it has already been stated and proven more soundly by many authors. It is true we did not have positive peritoneal fluid culture because the setup of the hospital did not allow us to perform STI twice in the same patient, regardless it would have been from two collection sites. At the same time, this is the drive behind our case presentation – the strangeness of normally looking, feeling, normal anatomical course tubes, the presence of peritonitis, and the positive cervico-vaginal PCR.
  • We have ammended the case presentation to state more clearly the rationale behind the decision to operate. The patient’s abdominal pain intensified despite medical management and the peritoneal fluid quantity augmented (stained, puss-like fluid).

Reviewer: Clarity in Table 1: consider adding reference ranges to improve readability. 

Authors: Thank you for your suggestion, we have added them.

Reviewers: Language and style: the English language is generally good, but minor grammatical and stylistic corrections are recommended to improve fluency and precision.

Authors: Thank you for your obervation. We have re-read and reshaped much of the article, we hope it sounds better now and no mistakes have escaped our reading.

Reviewer: The similarity analysis indicates a 19% match: I encourage the authors to carefully review the segments highlighted by the plagiarism checker. In several parts, particularly within the introduction and discussion, the similarity may derive from previously published reviews or guidelines.

With these revisions, the manuscript will be strengthened and suitable for publication.

Authors: Thank you for poiting this out, we did not copy-paste from other sources into our work. I would say there is little literature on the subject and you end up using the same words. However, we have rewritten much of the manuscript and hope to be out of the range of suspicion.

the subject and you end up using the same words. However, we have rewritten much of the manuscript and hope to be out of the range of suspicion.

Reviewer 2 Report

Comments and Suggestions for Authors

General Comments: This case report describes a rare presentation of primary peritonitis in an immunocompetent young woman caused by polymicrobial sexually transmitted infections (STIs). The case is interesting as it highlights the potential for severe abdominal infection from genital pathogens even in the absence of classic macroscopic signs of Pelvic Inflammatory Disease (PID) during laparoscopy. However, there are significant issues with the presentation of data (specifically Figure 2) and terminology that need to be addressed before publication.

Major Comments:

  1. Figure 2 Quality and Language: Figure 2 is currently a screenshot of a raw laboratory report in a non-English language (presumably Romanian, e.g., "VALORI BIOLOGICE DE REFERINŢĂ", "Pozitiv"). This is not suitable for an international English-language journal. The authors must replace this image with a properly formatted Table in English that summarizes the pathogens tested, the result (Positive/Negative), and the method used (NAAT).

  2. Terminology (Primitive vs. Primary): The title uses the term "Primitive peritonitis" , while the text predominantly uses "Primary peritonitis". "Primary peritonitis" is the standard medical term in English literature. Please standardize the terminology to "Primary peritonitis" throughout the manuscript, including the title.

  3. Diagnosis Clarification (PID vs. Primary Peritonitis): There is a contradiction in the text regarding the diagnosis of PID.

    • In the Case Presentation, it is stated: "an initial diagnosis of uncomplicated pelvic inflammatory disease (PID) was established".

    • However, in the Discussion, it is stated: "subsequent clinical and imaging evaluations excluded PID as the underlying cause".

    • While the authors argue this is "primary peritonitis" because the tubes appeared normal, the presence of C. trachomatis and M. hominis in the genital tract strongly suggests an ascending infection (which is the pathophysiology of PID). The fact that the tubes were not macroscopically purulent does not fully exclude "subclinical" salpingitis leading to peritonitis. The authors should discuss this nuance. It might be more accurate to describe this as "Peritonitis secondary to ascending STI infection without macroscopic salpingitis" rather than true "Primary Peritonitis" (which is typically monomicrobial and hematogenous, e.g., SBP in cirrhosis).

  4. Microbiological Confirmation: The authors state that bacteriological samples from the peritoneum were negative. Did the authors perform PCR/NAAT specifically on the peritoneal fluid? If the STI pathogens were only found in the vaginal/cervical swabs and not confirmed in the abdomen, the causal link is presumptive (though highly likely). This limitation should be explicitly stated. If peritoneal PCR was done, please include those results.

Minor Comments:

  1. English Phrasing:

    • Line 67: "The vaginal tact objectified pain...". "Vaginal tact" is a direct translation from Romance languages (e.g., tacto vaginal). In English, this should be "Bimanual pelvic examination revealed...".

    • Line 61: "VAS (visual analogic scale)" should be "Visual Analog Scale".

    • Abstract: "Primitive peritonitis" -> "Primary peritonitis".

  2. Antibiotic Logic: The authors mention that Mycoplasma hominis is intrinsically resistant to macrolides and beta-lactams and requires tetracyclines. However, the discharge prescription included "Amoxicillin-clavulanate" alongside Doxycycline. While Doxycycline covers the atypical pathogens, what was the rationale for continuing Amoxicillin-clavulanate if the culture was negative and the pathogens were identified as STIs? Was it to cover potential gut flora translocation despite the negative culture? Please clarify.

  3. Table 1: The formatting of Table 1 in the PDF is slightly messy. Ensure that units (e.g., /µL, mg/dL) are clearly aligned with the values.

Author Response

Esteemed Reviewer,

Thank you very much for the time and effort invested into reading our manuscript and providing such insightful comments, upon reading we were stricken by our lack of clarity regarding some of the aspects you mentioned. We have done our best to improve the manuscript accoding to your suggestions and we hope you will find it amenable for publication in its updated format. All changes brought to the manuscript are highlighted in blue.

Major Comments:

Reviwer: Figure 2 Quality and Language: Figure 2 is currently a screenshot of a raw laboratory report in a non-English language (presumably Romanian, e.g., "VALORI BIOLOGICE DE REFERINŢĂ", "Pozitiv"). This is not suitable for an international English-language journal. The authors must replace this image with a properly formatted Table in English that summarizes the pathogens tested, the result (Positive/Negative), and the method used (NAAT).

Authors: Thank you for your comment, we have changed the results presentation to a table. We must have been too anxious to proove our result therefore we used the exact document from the lab, disregarding that it was written in Romanian. We apologize.

Reviewer: Terminology (Primitive vs. Primary): The title uses the term "Primitive peritonitis" , while the text predominantly uses "Primary peritonitis". "Primary peritonitis" is the standard medical term in English literature. Please standardize the terminology to "Primary peritonitis" throughout the manuscript, including the title.

Authors: We have amended the title and the one other use of “primitive” to “primary”. This was also a reminiscence from the Romanian translation, to confess. Also, with respect to your next comment, the entitle case structure was re-designed and the title was adjusted accordingly.

Reviewer: Diagnosis Clarification (PID vs. Primary Peritonitis): There is a contradiction in the text regarding the diagnosis of PID.

    • In the Case Presentation, it is stated: "an initial diagnosis of uncomplicated pelvic inflammatory disease (PID) was established".
    • However, in the Discussion, it is stated: "subsequent clinical and imaging evaluations excluded PID as the underlying cause".
    • While the authors argue this is "primary peritonitis" because the tubes appeared normal, the presence of C. trachomatis and M. hominis in the genital tract strongly suggests an ascending infection (which is the pathophysiology of PID). The fact that the tubes were not macroscopically purulent does not fully exclude "subclinical" salpingitis leading to peritonitis. The authors should discuss this nuance. It might be more accurate to describe this as "Peritonitis secondary to ascending STI infection without macroscopic salpingitis" rather than true "Primary Peritonitis" (which is typically monomicrobial and hematogenous, e.g., SBP in cirrhosis).

Authors: Thank you for your comment, it rocked our case building a bit. However, there are two angles to look at the situation and I hope you will agree they are both valid.

On the one hand, this case could be interpreted as symptomatic, acute PID with peritonitis, the diagnosis of PID being ascertained by the presence of cervical NAAT results, since it is true that mascoscopically normally looking salpinges do not rule out endoluminal salpingitis. However, it is also true that PID involves peritonitis more frequently in the context of complicated pyosalpinx or tubo-ovarian abcesses.

On the other hand, primary peritonitis, as a purulent intra-abdominal collection with no obvious cause, can be blamed, as repeatedly described in literature, on the dissemination from the genital tract.

Therefore, we could say these views are the two sides of the same coin.

Accordingly, we have adjusted course and reshaped the manuscript to depict this ambivalent situation.

We have rephrased the title to “Polymicrobial PID presenting as primary peritonitis in a young immunocompetent patient – case report and disease perspectives”.

Reviewer: Microbiological Confirmation: The authors state that bacteriological samples from the peritoneum were negative. Did the authors perform PCR/NAAT specifically on the peritoneal fluid? If the STI pathogens were only found in the vaginal/cervical swabs and not confirmed in the abdomen, the causal link is presumptive (though highly likely). This limitation should be explicitly stated. If peritoneal PCR was done, please include those results.
Authors: Thank you for your observation. Unfortunately our hospital does not provide PCR/NAAT at all, from any sampling site. The cervical swabs were sent to a private lab. In this context, we had the limitation of not being able to also request them from the peritoneal aspirate. We could only ask for culture on the classic microbiologic plates which are known to result negative. We have appended the text with this information.

Minor Comments:

Reviewer: English Phrasing:

    • Line 67: "The vaginal tact objectified pain...". "Vaginal tact" is a direct translation from Romance languages (e.g., tacto vaginal). In English, this should be "Bimanual pelvic examination revealed...".
    • Line 61: "VAS (visual analogic scale)" should be "Visual Analog Scale".
    • Abstract: "Primitive peritonitis" -> "Primary peritonitis".

Authors: Thank you for your comments, we have addresed all mistakes and they have been corrected.

Reviewer: Antibiotic Logic: The authors mention that Mycoplasma hominis is intrinsically resistant to macrolides and beta-lactams and requires tetracyclines. However, the discharge prescription included "Amoxicillin-clavulanate" alongside Doxycycline. While Doxycycline covers the atypical pathogens, what was the rationale for continuing Amoxicillin-clavulanate if the culture was negative and the pathogens were identified as STIs? Was it to cover potential gut flora translocation despite the negative culture? Please clarify.

Authors: Thank you for your comment. We started antibiotherapy empirically and at discharge we provided at step-down, oral regimen while maintaining the antimicrobial profile and the need to cover for gut flora translocation despite the negative cultures. We have included this information in the manuscript, in the case presentation.

Reviewer: Table 1: The formatting of Table 1 in the PDF is slightly messy. Ensure that units (e.g., /µL, mg/dL) are clearly aligned with the values.

Authors: Thank you. We have rechecked and corrected the table. We have also added reference values at the suggestion of another reviewer.

Round 2

Reviewer 2 Report

Comments and Suggestions for Authors

The authors have satisfactorily addressed the major concerns raised in the previous review round. The replacement of the original Figure 2 with a clear English table is appreciated, and the revised title ("Polymicrobial PID presenting as primary peritonitis...") accurately reflects the clinical scenario described. The discussion regarding the pathophysiology—ascending infection presenting clinically as primary peritonitis due to the absence of macroscopic tubal damage—is now much clearer.

However, there are a few minor but critical technical errors (typos in pathogen names and incorrect table numbering) that must be corrected before publication.

Specific Comments:

  1. Typos in Table 1: In the newly created Table 1 (Patient STI infection panel result), there are significant spelling errors in the pathogen names that need correction:

    • "lisichomonas vaginalis" should be "Trichomonas vaginalis".

    • "Neisseria gonorhoeae" should be "Neisseria gonorrhoeae".

  2. Table Numbering Discrepancy: The numbering of the tables appears to be inconsistent/duplicated:

    • Table 1: Patient STI infection panel (Page 4)

    • Table 2: Blood test results (Page 5)

    • Table 2 (Duplicate): Microorganisms isolated in PID patients (Page 6). This should be re-labeled as Table 3.

    • Table 3: Primary peritonitis cases... (Page 8). This should be re-labeled as Table 4.

    • Please check all in-text citations (e.g., "Table 1", "Table 2") to ensure they correspond to the correct tables after re-numbering.

Author Response

Esteemed Reviewer,

First of all, thank you very much for your so quick reply and attentive read of our manuscript. We are ashamed of not having noticed the mistakes you outlined, despite repeated verifications. We have corrected all the errors identified.

And, again, thank you for the push and for helping us put the case and the manuscript in a more correct and scientific perspective.